# Don't Let It Hallucinate: Premise Verification via Retrieval-Augmented Logical Reasoning

**Yuehan Qin**                                                                        *yuehanqi@usc.edu*
*University of Southern California*

**Shawn Li**                                                                               *li.li02@usc.edu*
*University of Southern California*

**Yi Nian**                                                                                    *yinian@usc.edu*
*University of Southern California*

**Xinyan Velocity Yu**                                                              *xinyany@usc.edu*
*University of Southern California*

**Yue Zhao**[*]                                                                             *yue.z@usc.edu*
*University of Southern California*

**Xuezhe Ma**[*]                                                                       *xuezhema@usc.edu*
*University of Southern California*

**Reviewed on OpenReview:** *https://openreview.net/forum?id=BDxStRGWba*

## Abstract

Large language models (LLMs) have shown substantial capacity for generating fluent, contextually appropriate responses. However, they can produce hallucinated outputs, especially when a user query includes one or more *false premises*—claims that contradict established facts. Such premises can mislead LLMs into offering fabricated or misleading details. Existing approaches include pretraining, fine-tuning, and inference-time techniques that often rely on access to logits or address hallucinations after they occur. These methods tend to be computationally expensive, require extensive training data, or lack proactive mechanisms to prevent hallucination before generation, limiting their efficiency in real-time applications. We propose a retrieval-based framework that identifies and addresses false premises *before* generation. Our method first transforms a user's query into a logical representation, then applies retrieval-augmented generation (RAG) to assess the validity of each premise using factual sources. Finally, we incorporate the verification results into the LLM's prompt to maintain factual consistency in the final output. Experiments show that this approach effectively reduces hallucinations, improves factual accuracy, and does not require access to model logits or large-scale fine-tuning.

## 1 Introduction

Large Language Models (LLMs) generate fluid, context-aware responses but can produce hallucinations when prompted with queries that include hidden factual errors (Manakul et al., 2023; Zheng et al., 2023). These

---

[*]Corresponding authors.

errors, known as *false premises*, are statements in a user's question that conflict with real-world facts. Even when LLMs can store accurate information, they may trust the incorrect assumptions embedded in the query and generate misleading outputs (Yuan et al., 2024). This is especially problematic in sensitive applications such as finance or healthcare, where mistakes can cause serious harm (Pal et al., 2023).

Prior research distinguishes between *factuality* hallucinations, where the output conflicts with known facts, and *faithfulness* hallucinations, where the response diverges from the provided context or user instructions (Snyder et al., 2024a). We focus on factuality hallucinations, particularly those driven by incorrect assumptions (false premises). These errors are common among LLM outputs (Huang et al., 2025; Snyder et al., 2024b), where Fig. 1 presents an example of a question with a false premise and the resulting hallucination. These premise-driven errors are particularly insidious as they can appear factually sound while being fundamentally incorrect.

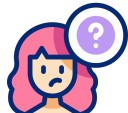

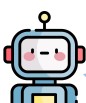

Is penicillin safe for treating my shellfish allergy symptoms?

Yes, penicillin is generally safe for most people and is one of the most widely prescribed antibiotics. It has a good safety profile when used appropriately. For shellfish allergy symptoms, penicillin can help reduce inflammation and prevent secondary bacterial infections that might occur from scratching or skin irritation …

Figure 1: LLM experiences factuality hallucination when faced with a false premise question, where both entities *shellfish allergy symptom* and *penicillin* exist but are not correctly aligned. The LLM's hallucinated response could delay life-saving treatment by incorrectly recommending antibiotics for allergic reactions.

Many methods attempt to address false premises after an LLM has already produced an answer. They include fine-tuning the model to detect invalid assumptions (Hu et al., 2023), applying contrastive decoding to surface inconsistencies (Shi et al., 2023; Chuang et al., 2024), and using uncertainty-based measures or logits to gauge inaccuracies (Pezeshkpour, 2023; Varshney et al., 2023). Although effective in some contexts, these approaches can be computationally demanding and do not necessarily prevent misinformation from appearing in the first place. Additionally, questions with false premises often maintain normal semantic flow, changing only a few tokens so that they are difficult to identify using traditional out-of-distribution detection (Vu et al., 2023). Even advanced LLMs can struggle with real-time truth evaluation, lacking the context or capacity to fully check every assumption (Hu et al., 2023; Liu et al., 2024c).

To address this challenge, we focus on *preventing* hallucinations rather than mitigating them post hoc. In our framework, we first transform the user's query into a logical form that highlights key entities or relations. We then employ retrieval-augmented generation (RAG) to check the accuracy of these statements against a knowledge graph. If contradictions are found, the query is flagged as containing a false premise prompting the model to correct or reject the assumption before formulating a final answer. This process, shown in Fig. 2, ensures that the LLM does not rely on erroneous details during response generation. By informing the LLM about any detected false premise in advance, we reduce the likelihood of hallucinations without requiring access to model logits or large-scale fine-tuning.

We summarize our contributions as follows:

**Logical Form Representation**: We first introduce logical forms to represent input queries and demonstrate their effectiveness across various types of graph retrievers. This logical approach enables accurate and systematic evaluation of statements provided in user prompts, particularly handling queries that may include false premises.

**Explicit False Premise Detection**: Our method improves the reliability of LLM-generated responses by explicitly detecting false premises and informing the LLM if a question contains a false premise.

**Hallucination Mitigation Without Output Generation or Model Logits**: Our approach reduces factual hallucinations without actual generation of responses or LLM logits and, therefore, can be seamlessly integrated into existing LLM frameworks and pipelines, offering a straightforward enhancement for improving factual accuracy.

## 2    Related Works

**False Premise**. A False Premise Question (FPQ) is a question containing incorrect facts that are not necessarily explicitly stated but might be mistakenly believed by the questioner (Yu et al., 2022; Kim et al., 2021). Recent studies (Yuan et al., 2024; Li et al., 2024; 2025c) have demonstrated that FPQs can induce factuality hallucination in LLMs, as they often respond directly to FPQs without verifying their validity. Notably, existing prompting techniques like few-shot prompting (Brown et al., 2020) and Chain-of-Thought (Wei et al., 2023), tend to increase hallucinations. Conversely, directly prompting LLMs to detect false premises degrades their performance on questions containing valid premises (Vu et al., 2023).

**Logical Forms**. Symbolic solvers and logical forms are applied to logical reasoning by grounding natural language in symbolic representations. The latest trend is integrating LLMs with symbolic solvers to enhance their performance Olausson et al. (2023); Pan et al. (2023a); Li et al. (2025b; 2023), where natural language is translated into symbolic logic forms and deterministic symbolic solvers are employed for inference, enabling more accurate logical problem-solving. Similarly, SymbCoT Xu et al. (2024) converts input text into symbolic formats such as first-order logic, generates reasoning plans through logical rule application, and verifies the reasoning process to ensure consistency. These methods demonstrate that incorporating symbolic improves the reliability and interpretability of LLM outputs, making them well-suited for tasks requiring logical consistency.

**Knowledge Graph Fact Checking and Question Answering**. In fact checking, RAG approaches verify data accuracy, with knowledge graph-driven RAG gaining attention for effectively leveraging structured knowledge. Recent works include: 1) *prompt-based* methods where (Pan et al., 2023b) evaluates evidence sufficiency and generates verification questions, and (Sun et al., 2024) performs hop-by-hop fact retrieval; 2) *graph-based* approaches where (He et al., 2024) formulates RAG as a Prize-Collecting Steiner Tree problem for subgraph extraction, and (Mavromatis & Karypis, 2024) uses graph neural networks for dense subgraph reasoning and answer retrieval; 3) *training-based* methods where (Zheng et al., 2024) develops dual encoders for query and subgraph evidence embedding, and (Liu et al., 2024a) trains encoders for retrieval and ranking processes, though requiring entity presence in the knowledge graph and relying on prompt-generated training data.

**Hallucination Mitigation**. Sources of LLM hallucinations originate from different stages in the LLM life cycle (Zhang et al., 2023a; Li et al., 2026; Shawn et al., 2025), leading existing mitigation methods to target specific stages: 1) *Pre-training*: Enhancing factual reliability by emphasizing credible texts, either by up-sampling trustworthy documents (Touvron et al., 2023) or prepending factual sentences with topic prefixes (Lee et al., 2023). 2) *Supervised Fine-tuning*: Curating high-quality, instruction-oriented datasets (Chen et al., 2024; Cao et al., 2024) improves factual accuracy more effectively than fine-tuning on unfiltered data, and remains more feasible compared to extensive pre-training. 3) *Reinforcement Learning from Human Feedback*: Aligning closely with human preferences may inadvertently encourage hallucinations or biased outputs, especially when instructions surpass the model's existing knowledge (Radhakrishnan et al., 2023; Wei et al., 2024). 4) *Inference*: Known as hallucination snowballing (Zhang et al., 2023b), LLMs occasionally magnify initial generation mistakes. Proposed inference-time solutions include new decoding strategies (Shi et al., 2023; Chuang et al., 2024), uncertainty analysis of model outputs (Xu & Ma, 2025; Liu et al., 2024b; Dhuliawala et al., 2023; Li et al., 2025a). However, these approaches either act post-hallucination or require access to model logits, thus being inefficient due to repeated prompting or limited to white-box LLM scenarios. We briefly discuss the comparison between our work and previous post-hoc hallucination mitigation method in Tab. 1 and detailed discussion can be found in Discussion and Appendix § A.4.

| Method | Training cost | Number of tokens | Training time | Model agnostic | Black-box Compatible |
|---|---|---|---|---|---|
| Post-hoc method | Depends on fine-tuning | Train: original query + answer Inference: original query | Depends on fine-tuning | No | No |
| Ours | Zero | Original query + logical form | Zero | Yes | Yes |

Table 1: Comparison of training and compatibility between post-hoc method and our method.

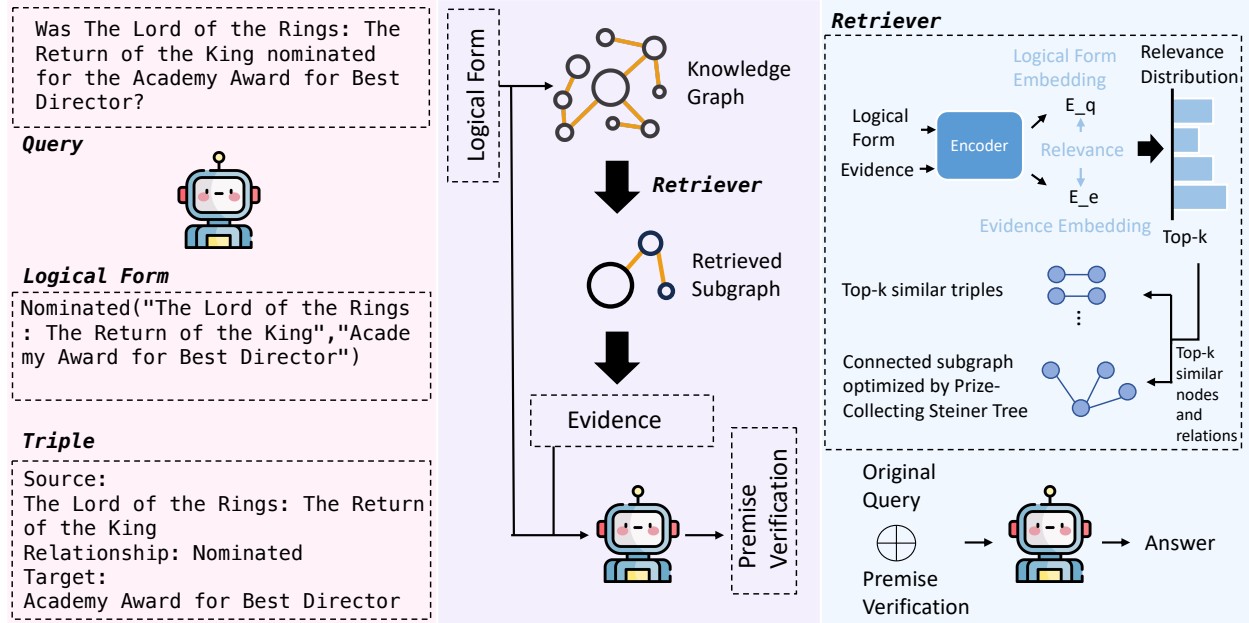

Figure 2: Overview of our approach. Left: The original query is converted into a logical form. Middle: The logical form is used to retrieve relevant elements from the knowledge graph and detect false premises. Right: Comparison of studied retrievers for aligning logical form with the knowledge graph. The LLM generates responses with reduced hallucination given prompts with premise verification.

## 3    Methodology

LLM hallucinations often stem from false premises in user queries. Instead of addressing hallucinations post-generation, we aim to prevent them by detecting and informing the presence of false premises to LLMs before response generation. Our proposed method achieves this through three key steps:

**Logical Form Conversion**. By converting the user query into a structured logical form representation, we extract its core meaning, making it easier to analyze its factual consistency. We demonstrate its effectiveness across various types of graph retrievers.

**Structured Retrieval and Verification**. Rather than relying solely on model-generated text, we retrieve external evidence to assess whether false premises exist in the query.

**Factual Consistency Enforcement**. The verified information is then incorporated into the LLM prompt, ensuring that the model generates responses aligned with factual data.

Our proposed method applies to knowledge graphs and datasets compatible with graph structures. We show the pseudocode summary of our approach in Algorithm 1.

---
**Algorithm 1** False premise detection and hallucination mitigation

---
**Input:** User query $q$, Knowledge graph $G$
**Output:** Hallucination mitigated response from LLM
1: Convert user query $q$ into logical representation $\mathcal{L}(q)$        ▷ (§3.2)
2: Extract logical assertions $P(x_1, x_2, \ldots, x_n)$ from $\mathcal{L}(q)$
3: Initialize maximum similarity score $Sim_{max} \leftarrow -\infty$        ▷ (§3.3)
4: Initialize optimal graph $G^* \leftarrow \emptyset$
5: Candidate set $G^* \leftarrow$ subsets of relevant subgraphs from $G$, i.e., $R(G)$
6: **for** triple $G' \in G$ **do**
7:      **if** retriever is embedding-based **then**
8:          Compute similarity via embeddings:
$$Sim \leftarrow \text{Sim}\left(\mathcal{L}(q), G'\right)$$
9:      **else if** retriever is non-parametric **then**
10:         Compute similarity using tree search criteria:
$$Sim \leftarrow \text{PCST}\left(\mathcal{L}(q), G'\right)$$
11:      **else if** retriever is LLM-based **then**
12:         Compute similarity using LLM scoring:
$$Sim \leftarrow \text{LLMScore}\left(\mathcal{L}(q), G'\right)$$
13:      **end if**
14:      **if** $Sim > Sim_{max}$ **then**
15:         $Sim_{max} \leftarrow Sim$
16:         $G^* \leftarrow G'$
17:      **end if**
18: **end for**
19: Define false premise indicator function:        ▷ (§3.1)
$$F(q) = \begin{cases} 1, & \text{if } q \text{ conflicts with retrieved evidence } G^* = R(q, G^*) \\ 0, & \text{otherwise} \end{cases}$$
20: **if** $F(q) = 1$ **then**        ▷ (§3.4)
21:      Update query as:
$$q \leftarrow q + \text{" Note: This question contains a false premise."}$$
22: **end if**
23: Generate response from LLM using updated query $q$
24: **return** Hallucination mitigated response from LLM

---

## 3.1 Problem Definition

**False Premise Detection**: Given a user query $q$, the function $F(q)$ determining whether $q$ contains a false premise can be defined as:

$$F(q) = \begin{cases} 1, & \text{if } q \text{ conflicts with retrieved evidence } R(q, G), \\ 0, & \text{otherwise,} \end{cases} \tag{1}$$

where $R$ denotes the retrieval function that extracts relevant evidence from a knowledge graph $G$. The query $q$ is evaluated against $R(q, G)$, and if contradictions are found, $q$ is deemed to contain a false premise ($F(q) = 1$); otherwise, it is considered valid ($F(q) = 0$). In this study, the function $F$ is achieved by RAG using a retriever that leverages logical form and a knowledge graph.

## 3.2 Logical Form Extraction

**Logical Form**: A logical form is a structured representation of statements or queries expressed using symbolic logic. It provides a structured way to capture semantic relationships within sentences, enabling precise and systematic reasoning. Given a natural language sentence query $q$, its logical form $\mathcal{L}(q)$ can be represented as: $\mathcal{L}(q) = P(x_1, x_2, \ldots, x_n)$, where $P$ denotes a predicate or relation, and $x_1, x_2, \ldots, x_n$ are variables or constants representing entities or concepts extracted from $q$. For example, for the query:

*Was The Lord of the Rings: The Return of the King nominated for the Academy Award for Best Director?*

Its logical form is:

*Nominated("The Lord of the Rings: The Return of the King", "Academy Award for Best Director")*

Here, *"The Lord of the Rings: The Return of the King"* and *"Academy Award for Best Director"* are entities, while *Nominated* is the relation. GPT-4o-mini (OpenAI, 2024) is used for extracting logical forms from the queries. For an input query $q$, we first ask the LLM to generate the corresponding logical form $L(q)$. Then, we extract the source, relationship, and target from $L(q)$. The prompt for logical form conversion is included in Appendix § A.3.

To evaluate the quality of the logical form conversion, we ask two annotators to manually grade the generated logical forms using a three-point scale: 1 (do not match), 2 (partially match), and 3 (match). Each annotator grades 100 randomly sampled outputs. Across all 200 samples, the generated logical forms receive a score of 3 in all of the cases. More details are in Appendix §A.6.

### 3.3 Retrieval

Given a user query $q$ in natural language, the retrieval stage aims to extract the most relevant elements (e.g., entities, triplets, paths, subgraphs) from knowledge graphs, which can be formulated as:

$$
\begin{aligned}
G^* &= \text{Graph-Retriever}(q, G) \\
&= \arg \max_{G \subseteq R(G)} p_\theta(G \mid q, G) \\
&= \arg \max_{G \subseteq R(G)} \text{Sim}(q, G),
\end{aligned} \tag{2}
$$

where $G^*$ is the optimal retrieved graph elements, and $\text{Sim}(\cdot, \cdot)$ is a function that measures the semantic similarity between user queries and the graph data. $R(\cdot)$ represents a function to narrow down the search range of subgraphs, considering the efficiency.

After converting a user query $q$ into a logical form representation $L(q)$, the retriever encodes the logical form and the graph triples, searches through the knowledge graph $G$, and extracts the most relevant triple or subgraph, applying different selection criteria depending on the retriever used in our study. Therefore, formula 2 can be further formulated to:

$$
\begin{aligned}
G^* &= \text{Graph-Retriever}(\mathcal{L}(q), G) \\
&= \arg \max_{G \subseteq R(G)} p_\theta(G \mid \mathcal{L}(q), G) \\
&= \arg \max_{G \subseteq R(G)} \text{Sim}(\mathcal{L}(q), G).
\end{aligned} \tag{3}
$$

We employ the pre-trained encoder *all-roberta-large-v1*[1] to encode the logical form and graph triplets. The representation $L(q)$ is used in both the similarity-based retrieval process and the step where the LLM assesses whether the original query $q$ contains a false premise.

### 3.4 Hallucination Mitigation

For a given user query $q$, if the false premise identification function $F(q)$ detects a false premise ($F(q) = 1$), we update its original query $q$ by appending a note:

$$
q = \begin{cases} q + W, & \text{if } F(q) = 1, \\ q, & \text{otherwise,} \end{cases} \tag{4}
$$

where $W$ = "Note: This question contains a false premise.", and $q$ is the modified query that explicitly flags the presence of a false premise when detected. Once the original query is updated, we evaluate LLM's responses and measure the effectiveness of the ensuing hallucination mitigation.

---

[1]https://huggingface.co/sentence-transformers/all-roberta-large-v1

# 4 Experiments

## 4.1 Dataset

KG-FPQ (Zhu et al., 2024) is a dataset containing true and false premise questions that are constructed from the KoPL knowledge graph, a high-quality subset of Wikidata. TPQs are generated from true triplets, while FPQs are created by replacing objects in false triplets via string matching. We evaluate the discriminative task in the art domain, where LLMs answer Yes-No questions (e.g., Is Hercules a cast member of 'The Lord of the Rings: the Return of the King'?). Dataset details are in Appendix §A.1.

The CREAK dataset (Onoe et al., 2021) is a benchmark for commonsense reasoning about entity knowledge. Unlike prior datasets focused on general physical or social scenarios, CREAK targets inferences that combine factual knowledge about specific entities with commonsense reasoning. It contains 13,000 human-authored English claims about entities labeled as true or false, along with a small contrast set. Each example requires understanding both factual attributes and commonsense implications. Dataset details are in Appendix §A.2.

The FEVER dataset Thorne et al. (2018) consists of natural-language claims labeled as *Supported*, *Refuted*, or *Not Enough Information*. Evidence is retrieved from Wikipedia, which functions as a natural-language knowledge base. Claims are created by modifying Wikipedia sentences and are subsequently verified independently, without access to their original sources. FEVER is widely adopted as a standard benchmark for fact verification and claim validation.

## 4.2 Experiment Setting

Our approach mitigates hallucination through a two-step process: First, we detect false premises in the user query. Then, we use the result of false premise detection along with the original query when providing input to the LLM. We use both the KG-FPQ dataset and the CREAK dataset for evaluating the premise detection task, and we use the KG-FPQ dataset for the hallucination mitigation task, since CREAK dataset contains statements, not questions.

### 4.2.1 False Premise Detection with Logical Form

In the false premise detection task, we look at different retrievers with and without the use of logical forms. Logical forms are used in 1) the retrieval stage, where the logical form $\mathcal{L}(q)$ is encoded to find the most relevant elements from knowledge graph $G$, and 2) the false premise detection stage, where the logical form is passed as input along with the retrieved evidence to LLM to determine whether the query contains false premise. The prompt detail is in Appendix A.3. We evaluate the use of logical forms in three configurations: 1) applying logical forms in both the retrieval stage and false premise detection stage, 2) using logical forms for retrieval and employing the original query for false premise detection, and 3) utilizing the original query for both stages. We further analyze the role of different components in the logical form through ablation studies in Appendix A.4.2.

### 4.2.2 False Premise Detection Methods

We evaluate how logical form impacts retrieval for false premise detection across the following retrievers:

1) **Direct Claim**: We directly query the LLM to determine whether the given question contains a false premise. The model is prompted with: *Does the following question contain a false premise? Answer with 'Yes' or 'No' only.*

2) **Embedding-based Retriever**: ***with RAG*** selects the top-$k^2$ relevant triples from the knowledge graph based on the cosine similarity between the query embedding and the graph triple embedding.

3) **Non-parametric Retriever**: ***G-retriever*** (He et al., 2024) uses Prize-Collecting Steiner Tree algorithm for extracting relevant subgraph from the knowledge graph. It does not rely on a trained model with learnable parameters.

---

[2]This work focuses on top-1 selection.

4) **LLM-based Retriever**: *GraphRAG/ToG* (Edge et al., 2025; Sun et al., 2024) asks the LLM to generate a score between 0 and 100, indicating how helpful the generated answer is in answering the target question. The answers are sorted in descending order of helpfulness score and used to generate the final answer returned to the user.

5) **SAC3** Zhang et al. (2024) is a hallucination detection approach that identifies hallucinations by assessing semantic-aware cross-check consistency, which involves generating semantically equivalent question perturbations and performing cross-model response consistency verification. We include SAC3 as baseline for the KG-FPQ dataset.

We use GPT-4o-mini as the LLM in the false premise detection task. Additional evaluations on other LLMs are provided in Appendix §A.4.4. These retrievers are included because they enable retrieval without task-specific fine-tuning, making them more adaptable across different domains. Unlike training-based retrievers, which require labeled data and extensive computation, non-parametric retriever uses structured knowledge, embedding-based retriever utilizes pre-trained encoders to transform queries and knowledge into a shared vector space for efficient retrieval, and LLM-based retrieval leverages pre-trained language models' generalization abilities. This setup evaluates the impact of logical forms on retrieval efficiency without the overhead of model training.

**Metrics.** We evaluate the false premise detection task using TPR (true positive rate), TNR (true negative rate), FPR (false positive rate), FNR (false negative rate), F1 score, and accuracy of the model successfully identifying questions containing false premises or not. Here, a *positive* instance refers to a question that contains a false premise. Higher TPR indicates better detection of false premises.

| | Direct Claim | with RAG | G-retriever | GraphRAG/ToG |
|---|---|---|---|---|
| Original Query for Both Stages | | | | |
| True Positives (TP%) | 44.44 | 33.33 | 88.89 | 8.89 |
| True Negatives (TN%) | 73.33 | 80.00 | 86.67 | 93.33 |
| False Positives (FP%) | 26.67 | 20.00 | 13.33 | 6.67 |
| False Negatives (FN%) | 55.56 | 66.67 | 11.11 | 91.11 |
| F1 Score (%) | 59.70 | 48.78 | 87.89 | 16.16 |
| Accuracy (%) | 69.20 | 73.33 | 88.57 | 81.27 |
| Logical Form for Retrieval and Original Query for False Premise Detection | | | | |
| True Positives (TP%) | 44.44 | 37.78 | 82.22 | 8.89 |
| True Negatives (TN%) | 73.33 | 86.67 | 93.33 | 93.33 |
| False Positives (FP%) | 26.67 | 13.33 | 6.67 | 6.67 |
| False Negatives (FN%) | 55.56 | 62.22 | 17.78 | 91.11 |
| F1 Score (%) | 59.70 | 53.97 | 86.97 | 16.16 |
| Accuracy (%) | 69.20 | 79.69 | 83.81 | 81.27 |
| Logical Form for Both Stages | | | | |
| True Positives (TP%) | 44.44 | 60.00 | 94.44 | 8.89 |
| True Negatives (TN%) | 73.33 | 86.67 | 99.05 | 93.33 |
| False Positives (FP%) | 26.67 | 13.33 | 0.95 | 6.67 |
| False Negatives (FN%) | 55.56 | 40.00 | 5.56 | 91.11 |
| F1 Score (%) | 59.70 | **73.97** | **97.12** | 16.16 |
| Accuracy (%) | 69.20 | **82.86** | **95.24** | 81.27 |

Table 2: KG-FPQ dataset: comparison of performance metrics across different retrieval methods using logical forms and/or original queries.

### 4.2.3 Hallucination Mitigation Methods

Having used logical forms to improve query structuring and false premise detection, we wish to illustrate how our logical form-based method further reduces hallucinations. We consider the following methods as our hallucination mitigation baselines, which are all inference-time hallucination mitigation strategies that

| | Direct Claim | with RAG | G-retriever | GraphRAG/ToG |
|---|---|---|---|---|
| Original Query for Both Stages | | | | |
| True Positives (TP%) | 72.5 | 62.3 | 24.6 | 89.9 |
| True Negatives (TN%) | 89.7 | 86.8 | 92.6 | 91.2 |
| False Positives (FP%) | 10.3 | 13.2 | 7.4 | 8.8 |
| False Negatives (FN%) | 27.5 | 37.7 | 75.4 | 10.1 |
| F1 Score (%) | 79.4 | 71.1 | 37.4 | 90.5 |
| Accuracy (%) | 81.0 | 74.5 | 58.4 | 90.5 |
| Logical Form for Retrieval and Original Query for False Premise Detection | | | | |
| True Positives (TP%) | 72.5 | 76.8 | 36.2 | 89.9 |
| True Negatives (TN%) | 89.7 | 92.6 | 83.8 | 88.2 |
| False Positives (FP%) | 10.3 | 7.4 | 16.2 | 11.8 |
| False Negatives (FN%) | 27.5 | 23.2 | 63.8 | 10.1 |
| F1 Score (%) | 79.4 | 83.5 | 47.6 | 89.2 |
| Accuracy (%) | 81.0 | 84.7 | 59.9 | 89.1 |
| Logical Form for Both Stages | | | | |
| True Positives (TP%) | 72.5 | 88.4 | 92.8 | 92.8 |
| True Negatives (TN%) | 89.7 | 92.6 | 83.8 | 91.2 |
| False Positives (FP%) | 10.3 | 7.4 | 16.2 | 8.8 |
| False Negatives (FN%) | 27.5 | 11.6 | 7.2 | 7.2 |
| F1 Score (%) | 79.4 | **90.4** | **88.9** | **92.1** |
| Accuracy (%) | 81.0 | **90.5** | **88.3** | **92.0** |

Table 3: CREAK dataset: comparison of performance metrics across different retrieval methods using logical forms and/or original queries.

| | Direct | WRAG (with RAG) | | | GraphRAG/ToG |
|---|---|---|---|---|---|
| | | Orig | LF-Retr | LF-Both | |
| True Positive (TP%) | 18.8 | 96.1 | 94.1 | 94.1 | 88.2 |
| True Negative (TN%) | 98.5 | 61.2 | 83.7 | 83.7 | 83.7 |
| False Positive (FP%) | 1.5 | 38.8 | 16.3 | 16.3 | 16.3 |
| False Negative (FN%) | 81.2 | 3.9 | 5.9 | 5.9 | 11.8 |
| F1 Score (%) | 31.3 | 82.4 | 89.7 | 89.7 | 86.5 |
| Accuracy (%) | 58.4 | 79.0 | 89.0 | 89.0 | 86.0 |

Table 4: FEVER dataset: comparison of performance metrics across different retrieval methods using logical forms and/or original queries. Since g-retriever is a graph-based retriever and does not apply to non-graph data, we do not include it here.

| Method | Accuracy | Number of tokens | Running time* | Model agnostic | Black-box Compatible |
|---|---|---|---|---|---|
| Contrastive Decoding | 84.8 | Original Query + Reasoning Step (Length $\gg$ Logical Form) | Context Retrieval Time + 10.6s | Agnostic to White Box Models | No |
| Our Method | 89.5 | Original Query + Logical Form | Context Retrieval Time + 0.6s | Yes | Yes |

Table 5: Comparison of performance and efficiency between contrastive decoding and our method on the KG-FPQ dataset. *Average running time of each query on NVIDIA RTX A6000 GPU using Llama-3.1-8B Instruct model. Both methods require context retrieval.

do not require access to logits or internal model weights that operate exclusively at the input level, ensuring a fair comparison:

1) **DirectAsk**: Directly query the LLMs for an answer without additional processing or external retrieval. This approach relies on the model's internal knowledge and reasoning capabilities to handle potential false premises.

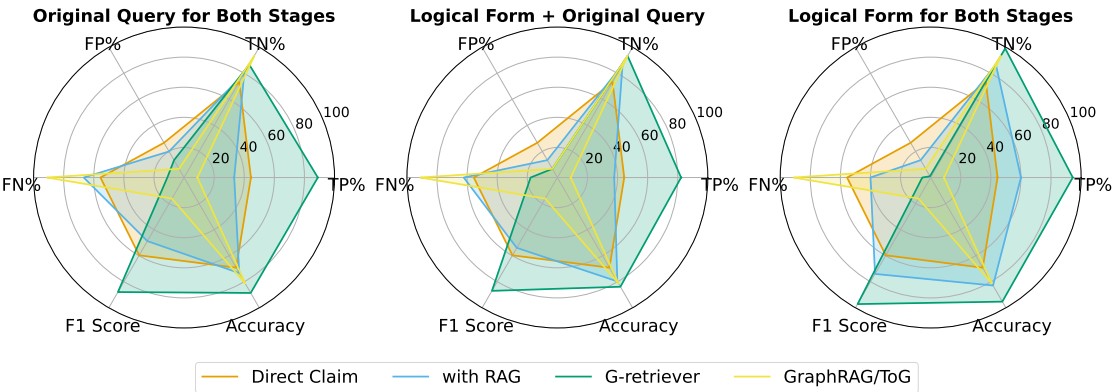

Figure 3: KG-FPQ dataset: comparison of performance metrics across different retrieval methods using logical forms and/or original queries.

2) **Prompt**: We encourage the LLM to assess potential false premises before generating a response by appending the following prompt to the original query: *This question may contain a fasle premise. [query]*

3) **Majority Vote (MajVote)**: We prompt the LLM three times with the same prompt and select the most frequent response as the final answer. This method improves reliability by reducing the impact of any single erroneous or hallucinated response. from LLM.

4) **Perplexity AI**[3]: Utilizes a search engine to retrieve and incorporate real-time information from the web, enabling it to provide answers based on the latest available web data. We use the version powered by GPT-4-Omni.

5) **Direct RAG**: Retrieves relevant entities from the knowledge graph and provides them as context to the LLM alongside the original query. This approach augments the model's internal knowledge directly with external information to improve answer accuracy and grounding.

We report the performances of the following LLMs: GPT-4o-mini (OpenAI, 2024), GPT-3.5-turbo (OpenAI, 2023), Llama-3.1-8B-Instruct (et al., 2024), Mistral-7B-Instruct-v0.2 (Jiang et al., 2023), Qwen2.5-7B-Instruct (Qwen et al., 2025), and Qwen-1.5-7b-chat (Bai et al., 2023). To better understand where the performance gains come from, we provide a per-model breakdown of error corrections and premise-specific changes (false- vs. true-premise queries) in Appendix §A.4.1.

| Models | DirectAsk | Prompt | MajVote | DirectRAG | Ours |
|---|---|---|---|---|---|
| GPT-4o-mini | 83.8 | 92.4 | 86.7 | 90.5 | **92.4** |
| GPT-3.5 | 93.3 | 93.3 | 92.4 | 90.5 | **94.3** |
| LLama-3.1 | 86.7 | 86.7 | 89.5 | 88.6 | **89.5** |
| Mistral-7B | 87.6 | 86.7 | 87.6 | 71.4 | **89.5** |
| Qwen2.5 | 92.4 | 86.7 | 92.4 | 92.4 | **95.2** |
| Qwen1.5 | 89.5 | 90.5 | 90.5 | 82.9 | **91.4** |
| Perplexity AI | | | 91.4 | | |

Table 6: Comparison of accuracy (%) of different hallucination mitigation methods.

**Metrics**. We evaluate question-answering accuracy on the hallucination mitigation task. Accuracy is calculated by string matching the responses of LLMs: for TPQs, answering "Yes" is considered correct; for FPQs, answering "No" is considered correct.

---

[3]https://www.perplexity.ai

# 5    Discussion

We show the result of the false premise detection task in Tab.2, 3, and 4 for the KG-FPQ, CREAK and FEVER dataset, respectively. The SAC3 baseline result is shown in Appendix § A.4.5. Tab. 6 presents the hallucination mitigation result.

**Using logical forms helps better identify false premises in the questions**. As shown in Tab. 2, for all three retrievers, explicitly incorporating logical forms into both retrieval and false premise detection stages significantly improves the identification of false premises. Sole reliance on original queries, even though potentially yielding high accuracy, tends to neglect accurate false premise identification, underscoring the importance of utilizing structured logical forms for tasks prioritizing precise false premise detection.

For the KG-FPQ dataset, among different types of retrievers, when using logical forms in both the retrieval and false premise detection stages, the G-retriever method achieves the highest TPR at 94.44%, demonstrating a strong capability in accurately identifying questions containing false premises. Notably, this method also achieves the highest F1 score (97.12%), indicating an optimal balance between precision and recall. Although the ToG method exhibits the highest TNR of 93.33%, it significantly underperforms in TPR and overall F1 score (16.16%), suggesting limited effectiveness in correctly identifying false premises.

Notably, when original queries are used in either retrieval, false premise detection, or both stages, despite achieving reasonable accuracy (73.33% and 79.69%), with RAG method shows significantly lower TPR (33.33% and 37.78%) compared to the first configuration. This suggests that relying on original queries alone, or in combination with logical forms in only one stage for detection, can achieve high accuracy due to correctly identifying negatives, it is less effective at capturing false premises, which is the primary focus of our task.

Similarly, for the CREAK dataset, according to Tab. 3, using logical forms in both retrieval and detection stages consistently boosts performance versus operating on the original query. The gains are most pronounced for G-retriever: F1 score increases from 71.1% to 90.4% and accuracy from 74.5% to 90.5%, driven by a large increase in TPR (62.3% to 88.4%). GraphRAG/ToG already performs strongly with original queries, but still benefits from using logical forms, reaching the best overall scores (F1 92.1% and Acc 92.0%) with higher TPR (89.9% to 92.8%) while keeping FPR modest. The mixed configuration for retrieval mainly helps G-retriever but is less reliable than using logical forms in both stages, underscoring that structured queries plus explicit false premise detection are jointly necessary. Logical forms improves evidence targeting and makes contradictions salient to LLMs. Overall, the results indicate that logic-aware framework is crucial for converting strong retrieval into consistent end-to-end factuality.

According to Tab. 4, the proposed design also yields consistent performance gains on natural language knowledge base, indicating that incorporating logical forms in both the premise detection and LLM response stages remains effective when the knowledge source is less structured. This suggests that the benefits of logical-form guided reasoning stem from improved premise understanding and response control, rather than reliance on a specific knowledge representation.

**Explicitly detecting and informing LLMs false premise mitigates hallucination**, as demonstrated in Tab. 6. Our proposed method, which directly communicates the presence of false premises to the models, achieves the highest accuracy: 92.4% with GPT-4o-mini, 94.3% with GPT-3.5, 95.2% with Qwen2.5, and 91.4% with Qwen-1.5. This performance surpasses alternative approaches such as *Direct Ask*, *Prompt*, *Majority Vote*, *DirectRAG*, and *Perplexity AI*.

*Majority Vote* does not perform well, likely due to hallucination snowballing, where repeated querying amplifies errors rather than correcting them. Additionally, while the *Prompt* method warns the model about potential false premises, it does not specifically tell the LLM which one contains false premises, negatively impacts performance on questions with valid premises, causes unnecessary cautiousness and reduces the model's ability to provide direct and confident answers. Besides, *Perplexity AI* does not perform as well potentially because the query format does not align well with graph data, leading to suboptimal retrieval of relevant information for certain types of questions. These findings emphasize the importance of tailoring hallucination mitigation strategies to both the model's reasoning process and the nature of the queries.

*Direct RAG* retrieves and feeds raw evidence to LLMs without explicitly structuring the reasoning process or highlighting inconsistencies between the query and retrieved facts. As a result, the model may surface relevant but semantically unaligned information, leading to shallow retrieval-based responses rather than true logical verification. In contrast, logical-form RAG + explicit false premise signaling forms a structured reasoning process: it decomposes the claim into logical predicates and explicitly indicates when a premise conflicts with retrieved evidence. This guides LLMs to perform fact-level reasoning and contradiction handling, reducing overreliance on surface overlap and improving factual precision and interpretability.

Beyond hallucination mitigation, the proposed premise verification mechanism can be extended to sensitive or controversial topics, where unverified premises may amplify misinformation or harmful narratives. By explicitly detecting unsupported assumptions prior to response generation, the method offers a principled way to prevent models from uncritically engaging with inaccurate or inflammatory premises, enabling safer and more grounded interactions in high-risk domains. This suggests a broader role for retrieval-augmented logical reasoning as a lightweight safeguard for responsible deployment, especially in scenarios where factual grounding is essential before engaging in downstream reasoning or dialogue.

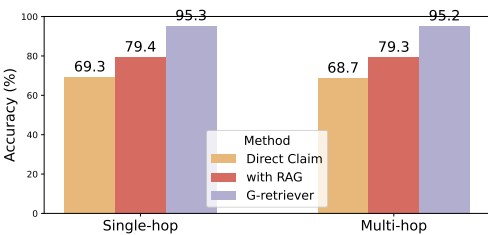

Figure 4: GPT-4o-mini and G-retriver: False premise detection accuracy across single-hop and multi-hop queries on the KG-FPQ dataset. Using logical form-based RAG mainly helps detect false premises in multi-hop questions.

**Our approach mostly improves false premise detection performance on multi-hop questions**, according to Fig. 4. The incorporation of logical form-based RAG leads to notable performance gains compared to direct claim evaluation. Specifically, while single-hop questions see moderate improvement, multi-hop questions benefit more, with false premise detection performance increasing from 68.7% in the direct claim setting to 79.3% with RAG and further to 95.2% when using the G-retriever. These results suggest that leveraging retrieval mechanisms enhances reasoning over multiple pieces of evidence, reinforcing the importance of retrieval-augmented methods for complex question-answering tasks. We present a case study to illustrate how our method improves performance on multi-hop questions in Appendix § A.4.6.

## 5.1 Computational Cost Analysis

In Tab. 5, we compare our method with the post-hoc Contrastive Decoding (Shi et al., 2023) approach in terms of computational efficiency and model compatibility (accuracy result based on Llama-3.1-8B). Our method reduces running time, uses fewer tokens by leveraging logical forms, and supports both model-agnostic and black-box settings. The proposed method introduces only a modest computational overhead, adding $\mathcal{O}(k+s)$ complexity to the retriever, where $k$ is the number of retrieved candidates and $s$ is the feature size used for logical-form reasoning and premise detection. In practice, this addition remains lightweight since most of the cost lies in the embedding model inference, which requires approximately 0.335 TFLOPs per example, while the remaining steps—retrieval, logical form conversion, and false-premise detection—are implemented as efficient API calls. In contrast, post-hoc methods rely on fine-tuning and lack general applicability across different model architectures. We also include performance comparison of Contrastive Decoding with other LLMs in Appendix § A.4.3.

### 5.1.1 Significance Test

To ensure that the reported performance differences are statistically meaningful, we conduct paired t-tests across model configurations. The results, summarized in Table 7 and Table 8 , confirm that the observed

improvements are statistically significant ($p < 0.05$) in most comparisons. Table 7 evaluates overall differences among performances of Direct Claim, with RAG, and G-Retriever on the KG-FPQ dataset. Table 8 focuses on intra-method variations, contrasting setups that use logical forms in both stages (GG), only one stage (GO), or none (OO). The results confirm that applying logical forms consistently across both retrieval and detection stages yields significantly higher performance, reinforcing the benefit of structured reasoning input.

| Comparison | p-value |
|---|---|
| G-Retriever – wRAG | 0.04 |
| Direct Claim – G-Retriever | 0.001 |
| wRAG – G-Retriever | 0.02 |

Table 7: Statistical significance comparison across retrieval methods.

| Comparison | p-value |
|---|---|
| G-Retriever (GG) – G-Retriever (GO) | $< 0.001$ |
| G-Retriever (GG) – G-Retriever (OO) | $< 0.001$ |
| wRAG (GG) – wRAG (GO) | 0.01 |

Table 8: Significance comparison of configurations within retrieval methods.

## 6 Conclusion

We propose a retrieval-augmented logical reasoning framework that detects false premises to mitigate LLM hallucinations. Our method explicitly detects and signals false premises, overcoming key limitations of current approaches that rely on model parameters or post-hoc corrections. By incorporating upfront false premise detection, we prevent hallucinations without requiring output generation or model logit access. Results show logical forms significantly improve false premise identification, especially for multi-hop reasoning questions. Our approach enhances LLM robustness by providing a structured mechanism to detect and handle misleading inputs before they influence downstream responses. This reinforces the importance of structured reasoning techniques in improving model reliability.

## Acknowledgments

This work was partially supported by the National Science Foundation under Award No. 2449280. Any opinions, findings, and conclusions or recommendations expressed are those of the authors and do not necessarily reflect the views of the National Science Foundation. The authors also gratefully acknowledge support from the Amazon Research Awards and Capital One Research Awards.

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

# A  Appendix

## A.1  KG-FPQ Dataset Details

In KoPL (Zhu et al., 2024), each entity is linked to a specific concept, such as *Leonardo da Vinci* being connected to the concept of an *artist*. The knowledge graph includes 794 distinct concepts, categorized into domains based on general knowledge, enabling domain-based entity classification. For the art domain, the authors of (Zhu et al., 2024) manually selected 33 relations, ensuring that each relation is relevant to its domain and informative, avoiding ambiguity. For example, the relation *artist* is linked to the Art domain, while *family* is more ambiguous and excluded. Table 9 shows the representative concepts, relations and subjects in the art domain of KG-FPQ. The dataset comprises 4969 questions in the discriminative task for the art domain, with each true premise question modified using the following editing methods: Neighbor-Same-Concept (NSC), Neighbor-Different-Concept (NDC), Not-Neighbor-Same-Concept (NNSC), Not-Neighbor-Different-Concept (NNDC), Not-Neighbor-Same-Relation (NNSR), and Not-Neighbor-Different-Relation (NNDR).

| Domain | Concept e.g. | Concept Qty | Subject e.g. | Subject Qty | Relation e.g. | Relation Qty |
|--------|-------------|-------------|--------------|-------------|---------------|--------------|
| Art | film television series drama | 44 | Titanic Modern Family Hamlet | 1754 | cast member composer narrative location | 33 |

Table 9: Representative concepts, relations, and subjects in KG-FPQ art domain.

## A.2  CREAK Dataset Details

The CREAK dataset Onoe et al. (2021) is designed to test whether language models can combine factual knowledge about specific entities with commonsense reasoning. It consists of 13k English claims covering 2.7k entities, each labeled as true or false. These claims require reasoning that bridges factual information (e.g., "Harry Potter is a wizard") with unstated commonsense inferences (e.g., "If someone is good at a skill, they can teach it"). Unlike prior commonsense benchmarks that focus on generic physical or social scenarios, CREAK emphasizes entity-grounded reasoning and assesses whether models can verify claims that depend on both knowledge retrieval and implicit reasoning. Table 10 summarizes the dataset statistics.

| Split | # Claims | | | Average Length | # Unique Entities | Vocab Size |
|-------|----------|------|-------|----------------|-------------------|------------|
| | Total | True | False | (# tokens) | | |
| Train | 10,176 | 5,088 | 5,088 | 10.8 | 2,096 | 19,006 |
| Dev | 1,371 | 691 | 680 | 9.7 | 531 | 4,520 |
| Test | 1,371 | 707 | 664 | 9.9 | 538 | 4,620 |
| Test (Contrast) | 500 | 250 | 250 | 10.0 | 226 | 1,596 |

Table 10: Data statistics of CREAK.

## A.3  Prompt Details

The following prompt is used to combine the information retrieved from the knowledge graph $G$ (context) and the query logical form $\mathcal{L}(q)$ (query) to form the input to the LLMs discussed in the Section *False Premise Detection with Logical Form*.

```
Given the context below, does the following question contain a false premise?  Answer
with 'Yes' or 'No' only.  Note that the context is provided as valid facts in a triple.
Context:  [context].  Query:  [query].
```

We use the following prompt for logical form conversion:

```
You are given a question.  The task is to:  1) define all the predicates used in the
question.  2) parse the question into logic rules based on the defined predicates 3)
translate any logical rules implied by the question.  4) convert the question into
```

```
a logical form using predicate logic.  Provide your final answer in the following
format:  Logical form:  Predicate1(entity1, entity2).  Keep all expressions concise and
consistent.  Use standard predicate logic notation.
```

## A.4  Additional Results

### A.4.1  Query-level Hallucination Mitigation Analysis

| Model | Extra Correct | FPQ Improved | TPQ Change |
|---|---|---|---|
| GPT-4o-mini | 427 | 392 | +35 |
| GPT-3.5 | 50 | 60 | -10 |
| Llama-3.1 | 139 | 115 | +24 |
| Mistral-7B | 94 | 108 | -14 |
| Qwen2.5 | 139 | 124 | +15 |
| Qwen1.5 | 94 | 74 | +20 |

Table 11: Premise-level breakdown of hallucination mitigation improvements. *Extra Correct* counts queries newly answered correctly compared to direct prompting. *FPQ Improved* counts false-premise queries corrected after false-premise detection. *TPQ Change* indicates net changes on true-premise queries after the detection and informing process.

### A.4.2  Logical Form Ablation Study

| | Full | w/o Rel. | w/o Ent$_1$ | w/o Ent$_2$ |
|---|---|---|---|---|
| Logical Form for Retrieval + Original Query for Detection | | | | |
| TPR | 0.377 | 0.044 | 0.067 | 0.011 |
| TNR | 0.866 | 0.000 | 0.067 | 0.267 |
| FPR | 0.133 | 1.000 | 0.933 | 0.733 |
| FNR | 0.622 | 0.956 | 0.933 | 0.989 |
| F1 | 0.540 | 0.073 | 0.109 | 0.020 |
| Acc | 0.800 | 0.038 | 0.067 | 0.048 |
| Logical Form for Both Stages | | | | |
| TPR | 0.600 | 0.044 | 0.067 | 0.044 |
| TNR | 0.866 | 0.000 | 0.133 | 0.267 |
| FPR | 0.133 | 1.000 | 0.867 | 0.733 |
| FNR | 0.400 | 0.956 | 0.933 | 0.956 |
| F1 | 0.740 | 0.073 | 0.109 | 0.076 |
| Acc | 0.829 | 0.038 | 0.076 | 0.076 |

Table 12: Logical form ablation results using **with RAG**.

We examine the contribution of individual components in the logical form, such as entity arguments and relational structure, by analyzing how the removal of specific elements affects false-premise detection across different retrieval backends. We consider variants where the logical form is used for retrieval while the original query is used for false-premise detection, as well as variants where the logical form is applied to both stages. The results are shown in Tab. 12, 13, 14. Across all retrieval backends, removing relational structure or entity arguments from the logical form consistently degrades performance. This suggests that the logical form is most effective when its constituent elements are jointly preserved, and that each component contributes complementary information for identifying unsupported premises.

### A.4.3  Comparison with Post-hoc Method

Tab. 18 presents a performance comparison between Contrastive Decoding (Shi et al., 2023), a post-hoc hallucination mitigation method, and other LLMs (Mistral-7B, Qwen1.5, Qwen2.5-7B-Instruct, Llama-3.1-8B-Instruct). Our method achieves improved performance over Contrastive Decoding on all models except Mistral-7B.

| | Full | w/o Rel. | w/o Ent$_1$ | w/o Ent$_2$ |
|---|---|---|---|---|
| Logical Form for Retrieval + Original Query for Detection | | | | |
| TPR | 0.822 | 0.178 | 0.078 | 0.111 |
| TNR | 0.933 | 0.133 | 0.067 | 0.133 |
| FPR | 0.066 | 0.867 | 0.933 | 0.867 |
| FNR | 0.177 | 0.822 | 0.922 | 0.889 |
| F1 | 0.869 | 0.270 | 0.124 | 0.177 |
| Acc | 0.838 | 0.171 | 0.076 | 0.114 |
| Logical Form for Both Stages | | | | |
| TPR | 0.944 | 0.167 | 0.089 | 0.133 |
| TNR | 0.991 | 0.133 | 0.133 | 0.200 |
| FPR | 0.009 | 0.867 | 0.867 | 0.800 |
| FNR | 0.056 | 0.833 | 0.911 | 0.867 |
| F1 | 0.971 | 0.254 | 0.144 | 0.211 |
| Acc | 0.952 | 0.162 | 0.095 | 0.143 |

Table 13: Logical form ablation results using **G-retriever**.

| | Full | w/o Rel. | w/o Ent$_1$ | w/o Ent$_2$ |
|---|---|---|---|---|
| Logical Form for Retrieval + Original Query for Detection | | | | |
| TPR | 0.089 | 0.056 | 0.089 | 0.078 |
| TNR | 0.933 | 0.133 | 0.200 | 0.267 |
| FPR | 0.067 | 0.867 | 0.800 | 0.733 |
| FNR | 0.911 | 0.944 | 0.911 | 0.922 |
| F1 | 0.162 | 0.093 | 0.145 | 0.130 |
| Acc | 0.813 | 0.067 | 0.105 | 0.105 |
| Logical Form for Both Stages | | | | |
| TPR | 0.089 | 0.100 | 0.100 | 0.067 |
| TNR | 0.933 | 0.133 | 0.200 | 0.267 |
| FPR | 0.067 | 0.867 | 0.800 | 0.733 |
| FNR | 0.911 | 0.900 | 0.900 | 0.933 |
| F1 | 0.162 | 0.163 | 0.162 | 0.112 |
| Acc | 0.813 | 0.104 | 0.114 | 0.095 |

Table 14: Logical form ablation results using **GraphRAG/ToG**.

### A.4.4 False Premise Detection

We additionally evaluate Llama-3.1-8b, as well as GPT-3.5-turbo and G-retriever on the false premise detection task using our method. The results are presented below (Tab. 15 and Tab. 17, Fig. 3). Notably, when original queries are used in either retrieval, false premise detection, or both stages, despite achieving high accuracy (91.11%), G-retriever shows a markedly lower TPR (37.78%) compared to the first configuration. This suggests that relying on original queries alone, or in combination with logical forms in only one stage for detection, can achieve high accuracy due to correctly identifying negatives, it is less effective at capturing false premises, which is the primary focus of our task.

### A.4.5 Premise Detection Baseline

We include SAC3 Zhang et al. (2024) as baseline for premise detection for the KG-FPQ dataset. Our proposed approach achieves better performance when considering both F1 score and Accuracy (see Table 2 for comparison).

### A.4.6 Case Study

We perform a case study demonstrating how our method enhances multi-hop question performance. We use the *with RAG* method example here, where the edited object is not directly connected to the subject within the KoPL knowledge graph.

| Metric | Direct Claim | with RAG | G-retriever | GraphRAG/ToG |
|---|---|---|---|---|
| Original Query for Both Stages | | | | |
| TPR | 0.878 | 0.811 | 0.644 | 0.800 |
| TNR | 0.200 | 0.467 | 0.467 | 0.267 |
| FPR | 0.800 | 0.533 | 0.533 | 0.733 |
| FNR | 0.122 | 0.189 | 0.356 | 0.200 |
| F1 | 0.873 | 0.854 | 0.744 | 0.832 |
| Accuracy | 0.781 | 0.762 | 0.619 | 0.724 |
| Logical Form for Retrieval and Original Query for False Premise Detection | | | | |
| TPR | 0.878 | 0.811 | 0.711 | 0.800 |
| TNR | 0.200 | 0.400 | 0.333 | 0.267 |
| FPR | 0.800 | 0.600 | 0.667 | 0.733 |
| FNR | 0.122 | 0.189 | 0.289 | 0.200 |
| F1 | 0.873 | 0.849 | 0.780 | 0.832 |
| Accuracy | 0.781 | 0.752 | 0.657 | 0.724 |
| Logical Form for Both Stages | | | | |
| TPR | 0.878 | 0.800 | 0.811 | 0.800 |
| TNR | 0.200 | 0.867 | 0.400 | 0.267 |
| FPR | 0.800 | 0.133 | 0.600 | 0.733 |
| FNR | 0.122 | 0.200 | 0.189 | 0.200 |
| F1 | 0.873 | 0.878 | 0.849 | 0.832 |
| Accuracy | 0.781 | 0.810 | 0.752 | 0.724 |

Table 15: KG-FPQ dataset: comparison of performance metrics across different retrieval methods using original queries and logical forms at different stages using Llama-3.1-8b.

| Method | TPR | TNR | FPR | FNR | F1 | Acc |
|---|---|---|---|---|---|---|
| SAC3 | 81.1 | 73.3 | 26.7 | 18.9 | 87.4 | 80.0 |

Table 16: Performance of SAC3 on premise detection on KG-FPQ dataset.

**Question**: Is "The Dark Knight" the recipient of the 16th Screen Actors Guild Awards?

**Answer**:      Yes, "The Dark Knight" won the award for Outstanding Performance by a Stunt Ensemble in a Motion Picture at the 16th Screen Actors Guild Awards.

**Logical Form**:  is a recipient of('The Dark Knight', 16th Screen Actors Guild Awards)

**Retrieved Graph Triple**:  ['The Dark Knight', 'award received', '81st Academy Awards']

After Detecting and Informing LLM of the Presence of a False Premise:

**Corrected Answer**:

```
No, "The Dark Knight" was not the recipient of the 16th Screen Actors Guild Awards.
That year's SAG Award for Outstanding Performance by a Cast in a Motion Picture went
to Inglourious Basterds, not The Dark Knight.
```

## A.5   Evaluation under Longer Queries

Longer questions often introduce additional contextual cues, compositional structures, and distracting details, which can obscure the core false assumption and make premise verification more challenging. Evaluating under such settings is important for understanding whether a model can robustly identify and reason about incorrect premises rather than relying on surface-level patterns.

As an initial step, we select the first 105 questions from the KG-FPQ dataset and use an LLM (gpt-4o-mini) to rewrite them into longer, more natural user queries while preserving the original (true or false) premises. We then evaluate our method on both false-premise detection and hallucination mitigation under this longer-context setting. The prompt used for query rewriting, along with the rewritten queries, is released

| Metric | G-retriever |
|---|---|
| Original Query for Both Stages | |
| True Positives (TP%) | 37.78 |
| True Negatives (TN%) | 100.00 |
| False Positives (FP%) | 0.00 |
| False Negatives (FN%) | 62.22 |
| F1 Score (%) | 54.84 |
| Accuracy (%) | 91.11 |
| Logical Form + Original Query | |
| True Positives (TP%) | 37.78 |
| True Negatives (TN%) | 100.00 |
| False Positives (FP%) | 0.00 |
| False Negatives (FN%) | 62.22 |
| F1 Score (%) | 54.84 |
| Accuracy (%) | 91.11 |
| Logical Form for Both Stages | |
| True Positives (TP%) | 75.56 |
| True Negatives (TN%) | 80.00 |
| False Positives (FP%) | 20.00 |
| False Negatives (FN%) | 24.44 |
| F1 Score (%) | 84.47 |
| Accuracy (%) | 79.37 |

Table 17: False Premise Detection Performance using GPT-3.5-turbo and G-retriever.

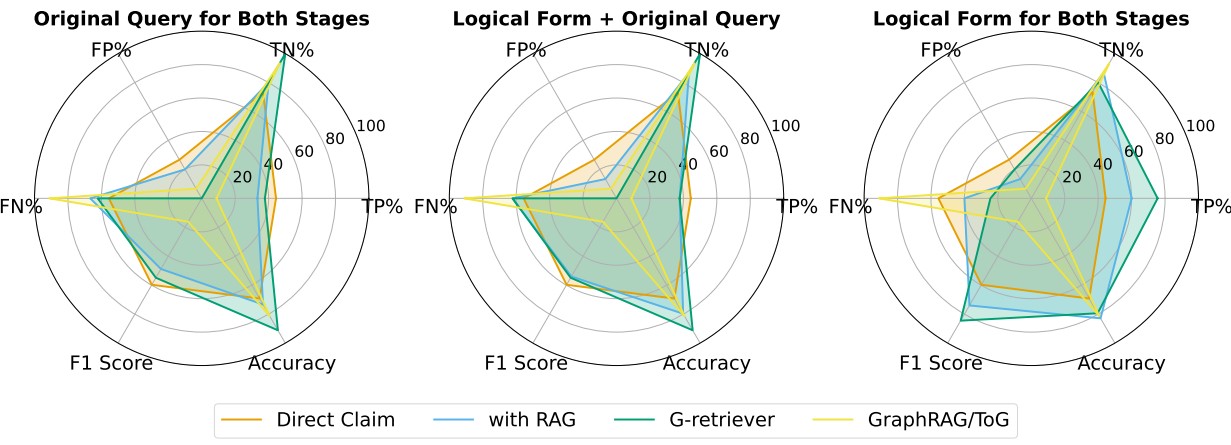

Figure 5: Additional comparison of performance metrics across different retrieval methods using logical forms and/or original queries.

in accompanying repository[4]. The False Premise Detection result and Hallucination Mitigation result are shown in Tab. 19 and Tab. 20. Overall, the results on this longer query subset show that the proposed method consistently improves performances on both false-premise detection and hallucination mitigation tasks, suggesting that its effectiveness extends beyond short or minimally phrased queries.

---

[4]https://github.com/yqin43/premise-verification

|  | Mistral-7B | Qwen1.5 | Qwen2.5-7B-Instruct | Llama-3.1-8B-Instruct |
|---|---|---|---|---|
| Contrastive Decoding | 89.5 | 76.2 | 85.7 | 84.8 |
| Ours | 87.6 | 89.5 | 92.4 | 86.7 |

Table 18: Comparison between contrastive decoding and our method across different LLMs. Note: GPT-3.5 and GPT-4o-mini are not included as logits are not available for contrastive decoding approach.

| Metric | Direct Claim | with RAG | G-retriever | GraphRAG/ToG |
|---|---|---|---|---|
| *Original Query for Both Stages* | | | | |
| TPR | 2.22 | 85.56 | 68.89 | 92.22 |
| TNR | 50.00 | 93.33 | 73.33 | 53.33 |
| FPR | 50.00 | 6.67 | 26.67 | 46.67 |
| FNR | 97.78 | 14.44 | 31.11 | 7.78 |
| F1 | 4.04 | 91.67 | 79.49 | 92.22 |
| Accuracy | 8.65 | 86.67 | 69.52 | 86.67 |
| *Logical Form for Retrieval and Original Query for Detection* | | | | |
| TPR | 2.22 | 94.44 | 93.33 | 92.22 |
| TNR | 50.00 | 73.33 | 53.33 | 53.33 |
| FPR | 50.00 | 26.67 | 46.67 | 46.67 |
| FNR | 97.78 | 5.56 | 6.67 | 7.78 |
| F1 | 4.04 | 94.97 | 92.82 | 92.22 |
| Accuracy | 8.65 | 91.43 | 87.62 | 86.67 |
| *Logical Form for Both Stages* | | | | |
| TPR | 2.22 | 94.44 | 97.78 | 92.22 |
| TNR | 50.00 | 73.33 | 46.67 | 53.33 |
| FPR | 50.00 | 26.67 | 53.33 | 46.67 |
| FNR | 97.78 | 5.56 | 2.22 | 7.78 |
| F1 | 4.04 | 94.97 | 94.62 | 92.22 |
| Accuracy | 8.65 | 91.43 | 90.48 | 86.67 |

Table 19: False-premise detection performance under longer queries.

### A.6 Logical Form Correctness Validation

Our current human evaluation was conducted by two Ph.D. students with relevant NLP/LLM research experience. All annotators reached full agreement, resulting in an inter-annotator agreement of 1.0.

### A.7 Additional Experiment Setup

All models are implemented and run on a multi-NVIDIA RTX 6000 Ada workstation. For *Logical Form Extraction* and *Retrieval*, we set parameters temperature $= 0$ and top_p $= 1$.

| Setting | Accuracy |
|---|---|
| Direct Ask | 0.933 |
| Ours | 0.952 |

Table 20: Hallucination mitigation accuracy under longer queries using GPT-4o-mini.

