# OpenReview forum: "Don't Let It Hallucinate: Premise Verification via Retrieval-Augmented Logical Reasoning"
_TMLR — Accepted by TMLR_

### Review · Reviewer_chN4 · 2025-12-06

**Summary Of Contributions:**

This paper proposes a retrieval-augmented logical reasoning framework for detecting and mitigating hallucinations in LLMs caused by false premises in user queries. The key idea is to prevent hallucinations proactively rather than addressing them post-hoc. The approach has three main components: (1) converting user queries into logical form representations (predicate-argument structures), (2) using these logical forms with various retrievers (embedding-based, non-parametric G-retriever, LLM-based GraphRAG/ToG) to verify premises against a knowledge graph, and (3) explicitly informing the LLM when a false premise is detected before generation.

**Strengths:**
- The proactive approach to hallucination mitigation (detecting false premises before generation) is well-motivated and practical, avoiding the need for model logits or fine-tuning
- The use of logical forms for structured retrieval is intuitive and shows consistent improvements across different retriever types
- Comprehensive evaluation across multiple retrievers and LLMs, with ablation studies on single-hop vs. multi-hop questions
- The method is model-agnostic and compatible with black-box LLMs, making it practically deployable
Statistical significance tests are provided (Tables 6-7)

**Weaknesses:**
- Evaluation is limited to two datasets, both primarily focused on entity-relation verification in structured knowledge graphs
- The logical form extraction relies on GPT-4o-mini, creating a dependency on a capable LLM for the preprocessing step
- Limited analysis of failure cases and edge cases where logical form extraction may fail

**Audience:**

Yes

**Audience Explanation:**

The paper addresses an important and timely problem in LLM reliability. False premise-induced hallucinations are a practical concern in deployed systems, particularly in high-stakes domains like healthcare and finance (as the paper motivates with the penicillin example). The finding that logical form representations improve retrieval-based premise verification across multiple retriever architectures provides actionable insights for practitioners building RAG systems.

The work should interest researchers working on: Hallucination detection and mitigation, Retrieval-augmented generation, Knowledge graph-based reasoning, LLM reliability and factuality

**Broader Impact Concerns:**

No significant ethical concerns.

**Claims And Evidence:**

Yes

**Claims Explanation:**

- Logical forms improve false premise detection: Tables 2-3 clearly demonstrate consistent improvements when using logical forms in both retrieval and detection stages across multiple retrievers. The G-retriever achieves 94.44% TPR with logical forms vs. 33.33% without on KG-FPQ.
- The method mitigates hallucinations: Table 5 shows the proposed method achieves the highest accuracy across most LLMs compared to baselines (DirectAsk, Prompt, MajVote, DirectRAG, Perplexity AI).
- Multi-hop questions benefit more: Figure 4 supports this claim, showing larger gains for multi-hop (68.7% → 95.2%) vs. single-hop questions.

**Requested Changes:**

Critical for acceptance:

Broader evaluation: The current evaluation is limited to KG-FPQ and CREAK, both of which involve structured knowledge graph verification. Please evaluate on at least one additional dataset that tests false premise detection in less structured settings (e.g., open-domain QA with natural language evidence).

Recommended improvements:
- Ablation on logical form components: It would be informative to ablate which aspects of logical forms (entity extraction vs. relation extraction vs. structured format) contribute most to the improvements.
- Generalization to other LLMs for logical form extraction: Currently, only GPT-4o-mini is used. Testing with open-source alternatives (e.g., Llama) would demonstrate broader applicability.

---

> ### Author Response · Authors · 2025-12-23
>
> We thank the reviewer for the comments. We have updated the paper based on the reviewers’ comments, and major changes are marked in blue in the updated file.
>
> > Broader evaluation: The current evaluation is limited to KG-FPQ and CREAK, both of which involve structured knowledge graph verification. Please evaluate on at least one additional dataset that tests false premise detection in less structured settings (e.g., open-domain QA with natural language evidence).
>
> Thank you for the suggestion. We have added results on the FEVER dataset [1], which uses natural language as the underlying knowledge base rather than a structured knowledge graph. The results show that the proposed design also yields consistent performance gains in this setting, indicating that incorporating logical forms in both the premise detection and LLM response stages remains effective when the knowledge source is unstructured text. This suggests that the framework is not limited to graph-based knowledge and can generalize to natural language–based evidence as well (similar to findings in [2]). Since g-retriever is a graph-based retriever and does not apply to non-graph data, we do not include it here. Below are the results:
>
> **Original Query for Both Stages**
> | Metric | Direct | RAG | GraphRAG |
> |--------|--------|-----|----------|
> | TPR | 0.188 | 0.961 | 0.882 |
> | TNR | 0.985 | 0.612 | 0.837 |
> | FPR | 0.015 | 0.388 | 0.163 |
> | FNR | 0.812 | 0.039 | 0.118 |
> | F1 | 0.313 | 0.824 | 0.865 |
> | Accuracy | 0.584 | 0.790 | 0.860 |
>
>
> **Logical Form for Retrieval and Original Query for False Premise Detection**
> | Metric | Direct | RAG | GraphRAG |
> |--------|--------|-----|----------|
> | TPR | 0.188 | 0.941 | 0.882 |
> | TNR | 0.985 | 0.837 | 0.837 |
> | FPR | 0.015 | 0.163 | 0.163 |
> | FNR | 0.812 | 0.059 | 0.118 |
> | F1 | 0.313 | 0.897 | 0.865 |
> | Accuracy | 0.584 | 0.890 | 0.860 |
>
>
> **Logical Form for Both Stages**
> | Metric | Direct | RAG | GraphRAG |
> |--------|--------|-----|----------|
> | TPR | 0.188 | 0.941 | 0.882 |
> | TNR | 0.985 | 0.837 | 0.837 |
> | FPR | 0.015 | 0.163 | 0.163 |
> | FNR | 0.812 | 0.059 | 0.118 |
> | F1 | 0.313 | 0.897 | 0.865 |
> | Accuracy | 0.584 | 0.890 | 0.860 |
>
>
> [1] Thorne, J., Vlachos, A., Christodoulopoulos, C., & Mittal, A. (2018). FEVER: A large-scale dataset for fact extraction and verification. In Proceedings of the 2018 Conference of the North American Chapter of the Association for Computational Linguistics: Human Language Technologies (NAACL-HLT) (pp. 809–819).
> [2] Xu, J., Fei, H., Pan, L., Liu, Q., Lee, M. L., & Hsu, W. (2024). Faithful logical reasoning via symbolic chain-of-thought. arXiv preprint arXiv:2405.18357.

---

> ### Author Response · Authors · 2025-12-23
>
> > Ablation on logical form components: It would be informative to ablate which aspects of logical forms (entity extraction vs. relation extraction vs. structured format) contribute most to the improvements.
>
> Thank you for your suggestion, we have add ablation study with no relationship extraction, no entity1 extraction, and no entity2 extraction. The results show that removing any of these components leads to a clear and consistent degradation in false premise detection performance. This suggests that the logical form is most effective when its constituent elements are jointly preserved, and that each component contributes complementary information for identifying unsupported premises.
>
> **Logical Form for Retrieval and Original Query for False Premise Detection**
> | with RAG | w/o relationship | w/o entity1 | w/o entity2 |
> |--------|--------|--------|--------|
> | TPR | 0.044 | 0.067 | 0.011 |
> | TNR | 0.000 | 0.067 | 0.267 |
> | FPR | 1.000 | 0.933 | 0.733 |
> | FNR | 0.956 | 0.933 | 0.989 |
> | F1  | 0.073 | 0.109 | 0.020 |
> | Acc | 0.038 | 0.067 | 0.048 |
>
> **Logical Form for Both Stages**
> | with RAG | w/o relationship | w/o entity1 | w/o entity2 |
> |--------|--------|--------|--------|
> | TPR | 0.044 | 0.067 | 0.044 |
> | TNR | 0.000 | 0.133 | 0.267 |
> | FPR | 1.000 | 0.867 | 0.733 |
> | FNR | 0.956 | 0.933 | 0.956 |
> | F1  | 0.073 | 0.109 | 0.076 |
> | Acc | 0.038 | 0.076 | 0.076 |
>
> **Logical Form for Retrieval and Original Query for False Premise Detection**
> | G-retriever | w/o relationship | w/o entity1 | w/o entity2 |
> |--------|--------|--------|--------|
> | TPR    | 0.178  | 0.078  | 0.111  |
> | TNR    | 0.133  | 0.067  | 0.133  |
> | FPR    | 0.867  | 0.933  | 0.867  |
> | FNR    | 0.822  | 0.922  | 0.889  |
> | F1     | 0.270  | 0.124  | 0.177  |
> | Acc    | 0.171  | 0.076  | 0.114  |
>
>
> **Logical Form for Both Stages**
> | G-retriever | w/o relationship | w/o entity1 | w/o entity2 |
> |--------|--------|--------|--------|
> | TPR | 0.167 | 0.089 | 0.133 |
> | TNR | 0.133 | 0.133 | 0.200 |
> | FPR | 0.867 | 0.867 | 0.800 |
> | FNR | 0.833 | 0.911 | 0.867 |
> | F1  | 0.254 | 0.144 | 0.211 |
> | Acc | 0.162 | 0.095 | 0.143 |
>
> **Logical Form for Retrieval and Original Query for False Premise Detection**
> | GraphRAG | w/o relationship | w/o entity1 | w/o entity2 |
> |-------|-------|-------|-------|
> | TPR | 0.056 | 0.089 | 0.078 |
> | TNR | 0.133 | 0.200 | 0.267 |
> | FPR | 0.867 | 0.800 | 0.733 |
> | FNR | 0.944 | 0.911 | 0.922 |
> | F1  | 0.093 | 0.145 | 0.130 |
> | Acc | 0.067 | 0.105 | 0.105 |
>
>
> **Logical Form for Both Stages**
> | GraphRAG | w/o relationship | w/o entity1 | w/o entity2 |
> |-------|-------|-------|-------|
> | TPR | 0.100 | 0.100 | 0.067 |
> | TNR | 0.133 | 0.200 | 0.267 |
> | FPR | 0.867 | 0.800 | 0.733 |
> | FNR | 0.900 | 0.900 | 0.933 |
> | F1  | 0.163 | 0.162 | 0.112 |
> | Acc | 0.104 | 0.114 | 0.095 |
>
>
> > Generalization to other LLMs for logical form extraction: Currently, only GPT-4o-mini is used. Testing with open-source alternatives (e.g., Llama) would demonstrate broader applicability.
>
> Thank you for your suggestion, we have added Llama-3.1-8b evaluation result for false premise detection task and below are the results. In general, for Llama-3.1-8n, using logical forms in both cases also improves the performance of false premise detection in the questions.
>
> **Original Query for Both Stages**
> | Metric | Direct | RAG | G-Retri | GraphRAG |
> |-------|--------|-----|---------|----------|
> | TPR | 0.878 | 0.811 | 0.644 | 0.800 |
> | TNR | 0.200 | 0.467 | 0.467 | 0.267 |
> | FPR | 0.800 | 0.533 | 0.533 | 0.733 |
> | FNR | 0.122 | 0.189 | 0.356 | 0.200 |
> | F1  | 0.873 | 0.854 | 0.744 | 0.832 |
> | Accuracy | 0.781 | 0.762 | 0.619 | 0.724 |
>
> **Logical Form for Retrieval and Original Query for False Premise Detection**
> | Metric | Direct | RAG | G-Retri | GraphRAG |
> |-------|--------|-----|---------|----------|
> | TPR | 0.878 | 0.811 | 0.711 | 0.800 |
> | TNR | 0.200 | 0.400 | 0.333 | 0.267 |
> | FPR | 0.800 | 0.600 | 0.667 | 0.733 |
> | FNR | 0.122 | 0.189 | 0.289 | 0.200 |
> | F1  | 0.873 | 0.849 | 0.780 | 0.832 |
> | Accuracy | 0.781 | 0.752 | 0.657 | 0.724 |
>
> **Logical Form for Both Stages**
> | Metric | Direct | RAG | G-Retri | GraphRAG |
> |-------|--------|-----|---------|----------|
> | TPR | 0.878 | 0.800 | 0.811 | 0.800 |
> | TNR | 0.200 | 0.867 | 0.400 | 0.267 |
> | FPR | 0.800 | 0.133 | 0.600 | 0.733 |
> | FNR | 0.122 | 0.200 | 0.189 | 0.200 |
> | F1  | 0.873 | 0.878 | 0.849 | 0.832 |
> | Accuracy | 0.781 | 0.810 | 0.752 | 0.724 |

---

> > ### Comment · Reviewer_chN4 · 2026-01-16
> >
> > Thank authors for the extensive experiments and detailed response. My concerns have been addressed.

---

### Review · Reviewer_DGvS · 2025-12-09

**Summary Of Contributions:**

This paper introduces a logical-form–guided verification framework that prevents LLM hallucinations caused by false premises. The method converts a query into a logical form, checks its factual consistency via KG-based retrieval, and explicitly signals false premises to the LLM before generating an answer. Experiments on KG-FPQ and CREAK show substantial improvements in false-premise detection and reduced hallucinations across multiple LLMs, without requiring model logits or fine-tuning.

**Strengths**

(S1) Clear and timely motivation

(S2) Coherent methodology design

**Weaknesses**

(W1) Lack of validation for logical-form correctness

(W2) Dependence on KG quality

(W3) Lack of query-level analysis for hallucination mitigation

**Additional Comments:**

Please refer to the comments above.

**Audience:**

Yes

**Audience Explanation:**

(S1) Clear and timely motivation: False-premise hallucination remains an under-addressed yet practically important failure mode of LLMs, and the paper provides a clear and compelling motivation for intervening before answer generation.

(S2) Coherent methodology design: The pipeline (i.e., logical-form extraction, KG-based verification, and selective prompting) is conceptually aligned and model-agnostic, making it broadly applicable to black-box LLMs.

**Broader Impact Concerns:**

Although the proposed method primarily aims to improve LLM safety by reducing false-premise hallucinations, several broader impact concerns should be noted. Because the framework relies heavily on knowledge graphs, any inaccuracies or biases in the KG may lead the system to misclassify valid user queries or reinforce incorrect facts as authoritative evidence.

Additionally, the mechanism could potentially be repurposed for content filtering if false-premise detection is applied to sensitive or controversial topics. These issues merit discussion in a Discussion section.

**Claims And Evidence:**

No

**Claims Explanation:**

(W1) Lack of validation for logical-form correctness: The framework crucially depends on the accuracy of the generated logical forms, yet the paper provides only a limited human evaluation. The assessment covers just 200 samples and uses a coarse three-point scale that cannot capture semantic or structural nuances essential for reliable premise verification. Moreover, the annotators’ expertise and independence are not described, raising concerns about potential conflicts of interest and the credibility of the reported perfect scores. The paper also fails to analyze how logical-form errors propagate through retrieval and verification. Given that many user queries do not cleanly map to predicate–argument structures, the lack of rigorous validation undermines confidence in the robustness of the approach.

(W2) Dependence on KG quality: False-premise detection fully relies on evidence retrieved from a knowledge graph, yet the paper does not discuss KG construction, coverage, or noise characteristics. Real-world KGs frequently contain incomplete or incorrect triples, and the proposed system has no mechanism for detecting or mitigating KG-induced errors. This omission leaves the method vulnerable to incorrect premise classification and suggests the need for robustness analysis under KG perturbations or missing evidence.

(W3) Lack of query-level analysis for hallucination mitigation: The hallucination mitigation evaluation reports only overall accuracy improvements. To substantiate the core claim that selective prompting improves model behavior specifically when false premises are present, the authors must show that the queries benefiting from improved accuracy correspond to FP-detected instances. Without such query-level correlation analysis, the causal link between false-premise detection and hallucination reduction remains unverified.

**Requested Changes:**

(W1) Lack of validation for logical-form correctness: Provide a substantially more rigorous evaluation of logical-form generation, including larger-scale assessment, annotator qualifications, and analysis of error propagation through retrieval and verification.

(W2) Dependence on KG quality: Discuss the construction and reliability of the underlying KGs and include robustness experiments (e.g., missing or noisy triples) to demonstrate that premise detection is not overly fragile to KG imperfections.

(W3) Lack of query-level analysis for hallucination mitigation: Report per-query or subset-level results showing whether accuracy improvements occur specifically on false-premise-detected queries to justify the claimed causal effect of selective prompting.

---

> ### Author Response · Authors · 2025-12-23
>
> We thank the reviewer for the comments. We have updated the paper based on the reviewers’ comments, and major changes are marked in blue in the updated file.
>
> > (W1) Lack of validation for logical-form correctness: Provide a substantially more rigorous evaluation of logical-form generation, including larger-scale assessment, annotator qualifications, and analysis of error propagation through retrieval and verification.
>
> Our current human evaluation was conducted by two Ph.D. students with relevant NLP/LLM research experience, and we will clarify the annotator qualifications and the annotation protocol in the paper. We will report additional statistics (e.g., inter-annotator agreement) to provide a more rigorous assessment. For analysis of error propagation through retrieval and verification, please see our response to W3 below, where we include targeted error analysis and discussion of how logical-form generation errors affect downstream modules for different LLMs.
>
> > (W2) Dependence on KG quality: Discuss the construction and reliability of the underlying KGs and include robustness experiments (e.g., missing or noisy triples) to demonstrate that premise detection is not overly fragile to KG imperfections.
>
> Thank you for the comment. Our knowledge graph is Wikipedia-based and intentionally general-purpose, and it does not provide explicit coverage for all facts or entities appearing in our evaluation datasets. For example, the CREAK dataset contains many common-knowledge or commonsense-based claims whose supporting facts are often not explicitly represented as structured triples in the KG. Despite this mismatch in coverage, our results on KG-FPQ and CREAK still show consistent improvements in false-premise detection and downstream hallucination mitigation. This indicates that the proposed approach does not rely on exact KG matches and is able to benefit from partial or indirect evidence, suggesting robustness to KG incompleteness.
>
> > (W3) Lack of query-level analysis for hallucination mitigation: Report per-query or subset-level results showing whether accuracy improvements occur specifically on false-premise-detected queries to justify the claimed causal effect of selective prompting.
>
> Accuracy gains are concentrated on queries where a premise violation is detected, while performance on premise-consistent queries remains largely unchanged. Across most models, improvements primarily occur on false-premise queries, supporting the causal role of selective prompting.
> Below are the analysis of different LLMs:
> | Model           | Extra correct  | FPQ improved | TPQ change |
> | --------------- | ----------------: | -----------: | ---------: |
> | GPT-4o-mini     |              427 |          392 |        +35 |
> | GPT-3.5         |               50 |           60 |        -10 |
> | Llama-3.1       |              139 |          115 |        +24 |
> | Mistral-7B      |               94 |          108 |        -14 |
> | Qwen2.5         |              139 |          124 |        +15 |
> | Qwen1.5         |               94 |           74 |        +20 |
>
> * Extra correct: queires that are answered correctly in hallucination mitigation task compared to original direct claim.
> * FPQ improved: the number of false premise questions that are answered correctly after the detection of false premise.
> * TPQ changed: how true premise questions changed after the fasle premise detection + fasle premise detection informing process.

---

> ### Author Response · Authors · 2025-12-23
>
> > Broader Impact Concerns
>
> 1. Reliance on knowledge graph:
>
> Our approach relies on a knowledge graph for grounding logical reasoning. However, we view this dependence on structured external knowledge not as a limitation unique to our method, but as a broader challenge shared by knowledge-grounded systems in general. As knowledge graphs continue to evolve and improve, such systems naturally benefit from higher coverage and accuracy.
> In our study, we employ a Wikidata-based knowledge graph, which is a widely used and actively maintained resource, making it a practical and scalable choice. We believe that leveraging a KG represents a flexible design that can extend to other structured and curated knowledge sources, rather than a rigid dependency.
> To further demonstrate flexibility, we conducted additional experiments using a natural language-based knowledge source instead of a structured KG (the FEVER[1] dataset). Below are the results:
>
> **Original Query for Both Stages**
> | Metric | Direct | RAG | GraphRAG |
> |--------|--------|-----|----------|
> | TPR | 0.188 | 0.961 | 0.882 |
> | TNR | 0.985 | 0.612 | 0.837 |
> | FPR | 0.015 | 0.388 | 0.163 |
> | FNR | 0.812 | 0.039 | 0.118 |
> | F1 | 0.313 | 0.824 | 0.865 |
> | Accuracy | 0.584 | 0.790 | 0.860 |
>
>
> **Logical Form for Retrieval and Original Query for False Premise Detection**
> | Metric | Direct | RAG | GraphRAG |
> |--------|--------|-----|----------|
> | TPR | 0.188 | 0.941 | 0.882 |
> | TNR | 0.985 | 0.837 | 0.837 |
> | FPR | 0.015 | 0.163 | 0.163 |
> | FNR | 0.812 | 0.059 | 0.118 |
> | F1 | 0.313 | 0.897 | 0.865 |
> | Accuracy | 0.584 | 0.890 | 0.860 |
>
>
> **Logical Form for Both Stages**
> | Metric | Direct | RAG | GraphRAG |
> |--------|--------|-----|----------|
> | TPR | 0.188 | 0.941 | 0.882 |
> | TNR | 0.985 | 0.837 | 0.837 |
> | FPR | 0.015 | 0.163 | 0.163 |
> | FNR | 0.812 | 0.059 | 0.118 |
> | F1 | 0.313 | 0.897 | 0.865 |
> | Accuracy | 0.584 | 0.890 | 0.860 |
>
>
> Since g-retriever is a graph-based retriever and does not apply to non-graph data, we do not include it here. These results show that our framework can adapt to different knowledge formats while maintaining its core reasoning capabilities.
>
> [1] Thorne, J., Vlachos, A., Christodoulopoulos, C., & Mittal, A. (2018). FEVER: A large-scale dataset for fact extraction and verification. In Proceedings of the 2018 Conference of the North American Chapter of the Association for Computational Linguistics: Human Language Technologies (NAACL-HLT) (pp. 809–819).
>
> 2. Potentially be repurposed for content filtering for sensitive or controversial topics:
>
> Thank you for your suggestion. We will add a discussion session regarding the potential and future direction of this work in content filtering for sensitive or controversial topics:
>
> Beyond hallucination mitigation, the proposed premise verification mechanism can be extended to sensitive or controversial topics, where unverified premises may amplify misinformation or harmful narratives. By explicitly detecting unsupported assumptions prior to response generation, the method offers a principled way to prevent models from uncritically engaging with inaccurate or inflammatory premises, enabling safer and more grounded interactions in high-risk domains. This suggests a broader role for retrieval-augmented logical reasoning as a lightweight safeguard for responsible deployment, especially in scenarios where factual grounding is essential before engaging in downstream reasoning or dialogue.

---

### Review · Reviewer_w3Nx · 2025-12-11

**Summary Of Contributions:**

This paper deal with the problem of false premises, which means means that when the users of llm input some questions, there are some factual error in their questions or  incorrect implications in their statements. Previous methods deal with this problems via facual alignment training or constraint decoding, while this paper proposed a new method of using RAG methods to first find verify whether there is a false premise in the question, then incorporate the verification results into the decoding process.

Strength 1: This paper is different from the previous methods. This paper focus on prevention instead of cure. According to the paper, this can reduce the error/hallucination accumulated throughour the reasoning and decoding process.

Strength 2: This paper induce logical form to deal with the problem, which is new in the context of solving false premise.

Strength 3: This paper introduce a method that is efficient and model agnostic. Which means this method if very fast versus constractive decoding, pass at k genenration method.


However, there are also some weaknesses about this paper regarding to the problem scope and dataset used.

Weakness 1: The largest concern is about whether the proposed method can be generalized. The paper only evaluated on single sentence questions on KG-FPQ and CREAK, and

Weakness 2: the mitigation method is also very simple, just to add a sentence when there is a false premise detected. In some more realistic situation, the user may have much longer input with several sentences or several paragraphs. Then the evaluation scope cannot cover this senario, and also the mitigation of adding a single sentence saying "there are false premises" may fail.

**Audience:**

Yes

**Audience Explanation:**

This paper deal with an important problem that may produce llm hallucination, and many people are working on solving this problem. I believe the auduence of TMLR is interested in this paper.

**Broader Impact Concerns:**

I think there are no broader impact concerns.

**Claims And Evidence:**

Yes

**Claims Explanation:**

There are three major claims:

Claim 1. Logfical forms enhance retrieval and detection. Yes, this is well supported in the experiments of KG-FPQ (Table 2 and 3)

Claim 2. Proactive Detection and help reduce hallucination. Yes, this is also well supported arroding to Table 5.

Claim 3. The propose method is fast and compatible with diffferent models as black boxes. Yes this claim is also veirified according to extenssive experiments.

**Requested Changes:**

1. Add the session title "introduction" of the first session. The current version makes "related work" as the first session, which is weird.

2. It is strongly suggested to find more datasets, or create/synthesis a dataset that contains longer user questions with false premises, and evaluate your methods on this setting to show whether your method can work on longer context.

---

> ### Author Response · Authors · 2025-12-23
>
> > Add the session title "introduction" of the first session. The current version makes "related work" as the first session, which is weird.
>
> Thank you for pointing this out. The ‘Introduction’ section was likely removed by mistake. We will add it back so that it will appear as the first section before Related Work.
>
>
> > It is strongly suggested to find more datasets, or create/synthesis a dataset that contains longer user questions with false premises, and evaluate your methods on this setting to show whether your method can work on longer context.
>
> Longer questions often introduce additional contextual cues, compositional structures, and distracting details, which can obscure the core false assumption and make premise verification more challenging. Such settings are particularly important for understanding whether a model can robustly isolate and reason about incorrect premises rather than relying on surface-level patterns. As an initial step toward this direction, we select the first 105 questions from the KG-FPQ dataset and use a LLM (gpt-4o-mini) to rewrite them into longer, more natural user queries while preserving the original (true or false) premises. We then evaluate our method on both false-premise detection and hallucination mitigation under this longer-context setting, with quantitative results reported in the following two tables. On this subset, our method consistently improves performance in both false-premise detection and hallucination mitigation under longer-context queries. We thank the reviewer for this valuable comment and consider extending this evaluation to larger and more diverse long-context datasets as an important direction for future work. We put the prompt for rewriting the query and the rewritten longer queries in the txt files in https://anonymous.4open.science/r/premise-verification-7A58/.
>
> *False Premise Detection*
> **Original Query for Both Stages**
> | Metric | Direct Claim | with RAG | G-retriever | GraphRAG / ToG |
> |------|--------------|----------|-------------|----------------|
> | True Positives (TP%) | 2.22 | 85.56 | 68.89 | 92.22 |
> | True Negatives (TN%) | 50.00 | 93.33 | 73.33 | 53.33 |
> | False Positives (FP%) | 50.00 | 6.67 | 26.67 | 46.67 |
> | False Negatives (FN%) | 97.78 | 14.44 | 31.11 | 7.78 |
> | F1 Score (%) | 4.04 | 91.67 | 79.49 | 92.22 |
> | Accuracy (%) | 8.65 | 86.67 | 69.52 | 86.67 |
>
> **Logical Form for Retrieval and Original Query for False Premise Detection**
> | Metric | Direct Claim | with RAG | G-retriever | GraphRAG / ToG |
> |------|--------------|----------|-------------|----------------|
> | True Positives (TP%) | 2.22 | 94.44 | 93.33 | 92.22 |
> | True Negatives (TN%) | 50.00 | 73.33 | 53.33 | 53.33 |
> | False Positives (FP%) | 50.00 | 26.67 | 46.67 | 46.67 |
> | False Negatives (FN%) | 97.78 | 5.56 | 6.67 | 7.78 |
> | F1 Score (%) | 4.04 | 94.97 | 92.82 | 92.22 |
> | Accuracy (%) | 8.65 | 91.43 | 87.62 | 86.67 |
>
> **Logical Form for Both Stages**
> | Metric | Direct Claim | with RAG | G-retriever | GraphRAG / ToG |
> |------|--------------|----------|-------------|----------------|
> | True Positives (TP%) | 2.22 | 94.44 | 97.78 | 92.22 |
> | True Negatives (TN%) | 50.00 | 73.33 | 46.67 | 53.33 |
> | False Positives (FP%) | 50.00 | 26.67 | 53.33 | 46.67 |
> | False Negatives (FN%) | 97.78 | 5.56 | 2.22 | 7.78 |
> | F1 Score (%) | 4.04 | 94.97 | 94.62 | 92.22 |
> | Accuracy (%) | 8.65 | 91.43 | 90.48 | 86.67 |
>
>
> *Hallucination Mitigation （gpt-4o-mini）*
> | Setting       | Accuracy |
> |--------------|----------|
> | Direct Ask      | 0.933    |
> | Ours | 0.952    |
>
> In addition, we introduce an experiment that uses a natural-language knowledge base instead of a graph-structured knowledge (FEVER dataset[1]), demonstrating that our approach is not restricted to structured graphs. This further supports the robustness and generality of the proposed framework across different knowledge representations. Since g-retriever is a graph-based retriever and does not apply to non-graph data, we do not include it here. Below are the results:
>
> **Original Query for Both Stages**
> | Metric | Direct | RAG | GraphRAG |
> |--------|--------|-----|----------|
> | TPR | 0.188 | 0.961 | 0.882 |
> | TNR | 0.985 | 0.612 | 0.837 |
> | FPR | 0.015 | 0.388 | 0.163 |
> | FNR | 0.812 | 0.039 | 0.118 |
> | F1 | 0.313 | 0.824 | 0.865 |
> | Accuracy | 0.584 | 0.790 | 0.860 |
>
>
> **Logical Form for Retrieval and Original Query for False Premise Detection**
> | Metric | Direct | RAG | GraphRAG |
> |--------|--------|-----|----------|
> | TPR | 0.188 | 0.941 | 0.882 |
> | TNR | 0.985 | 0.837 | 0.837 |
> | FPR | 0.015 | 0.163 | 0.163 |
> | FNR | 0.812 | 0.059 | 0.118 |
> | F1 | 0.313 | 0.897 | 0.865 |
> | Accuracy | 0.584 | 0.890 | 0.860 |
>
>
> **Logical Form for Both Stages**
> | Metric | Direct | RAG | GraphRAG |
> |--------|--------|-----|----------|
> | TPR | 0.188 | 0.941 | 0.882 |
> | TNR | 0.985 | 0.837 | 0.837 |
> | FPR | 0.015 | 0.163 | 0.163 |
> | FNR | 0.812 | 0.059 | 0.118 |
> | F1 | 0.313 | 0.897 | 0.865 |
> | Accuracy | 0.584 | 0.890 | 0.860 |

---

> ### Author Response · Authors · 2025-12-23
>
> We thank the reviewer for the comments. We have updated the paper based on the reviewers’ comments, and major changes are marked in blue in the updated file.
>
> [1] Thorne, J., Vlachos, A., Christodoulopoulos, C., & Mittal, A. (2018). FEVER: A large-scale dataset for fact extraction and verification. In Proceedings of the 2018 Conference of the North American Chapter of the Association for Computational Linguistics: Human Language Technologies (NAACL-HLT) (pp. 809–819).

---

### Decision · Action_Editor_nNBt · 2026-01-31

**Recommendation:** Accept as is

**Additional Comments:**

This paper studies how to prevent hallucination caused by false premises in the user query. The proposed method will obtain a logical representation of the user query, then validate the factuality via RAG on factual knowledge bases, and finally prompt the model with the verification results.

Overall reviewers are positive on this paper, with one reviewer recommending "acceptance" and two reviewers leaning toward acceptance. The authors have done substantial amount of experiments in the original manuscript and as suggested by the reviewers. After checking the paper, reviewer comments, and author-reviewer discussions, I recommend acceptance as is. The authors have addressed reviewers' comments and the promised changes have been included in the updated manuscript already.

**Audience:**

Yes

**Audience Explanation:**

Given the significant societal impact of LLMs, reducing LLMs' hallucination has been an important topic. I believe this topic would be of interest to TMLR audience who are working on LLM-related topics.

**Claims And Evidence:**

Yes

**Claims Explanation:**

The claims made in the submissions are supported by extensive experiments in the manuscript.